# Have Previous COVID-19 Vaccinations Shaped the Potential Enhancing Infection of Variant Strains?

**DOI:** 10.3390/vaccines12060567

**Published:** 2024-05-22

**Authors:** Husheng Xiong, Xiang Meng, Yanqin Song, Jiayi Zhong, Shuang Liu, Xun Zhu, Xin Ye, Yonghui Zhong, Dingmei Zhang

**Affiliations:** 1Department of Epidemiology, School of Public Health, Sun Yat-sen University, Guangzhou 510080, China; xionghsh5@mail2.sysu.edu.cn (H.X.); mengx8@mail2.sysu.edu.cn (X.M.); zhongjy58@mail2.sysu.edu.cn (J.Z.); liush333@mail2.sysu.edu.cn (S.L.); 2The Fourth People’s Hospital of Foshan City, Foshan 528000, China; syq830728@163.com; 3Department of Immunology and Microbiology, Zhongshan School of Medicine, Sun Yat-sen University, Guangzhou 510080, China; zhuxun8@mail.sysu.edu.cn; 4Tianhe District Center for Disease Control and Prevention, Guangzhou 510630, China; kfcusr888@163.com

**Keywords:** SARS-CoV-2, Omicron variant, vaccination, infection status

## Abstract

Objective: This study aimed to investigate the infection status of Omicron in the population and the association between COVID-19 vaccination and infection with Omicron. Methods: We conducted a cross-sectional study to openly recruit participants for a survey of SARS-CoV-2 infection by convenience sampling from 1 January to 15 January 2023 after a COVID-19 pandemic swept across China. Additionally, the binary logistic regression model was adopted to evaluate the association between COVID-19 vaccination and the infection outcomes or symptom severity, respectively. Meanwhile, the relations between the vaccination and duration of the symptoms were estimated via ordinal logistic analysis. Results: Of the 2007 participants, the prevalence of infection with Omicron was 82.9%. Compared with unvaccinated individuals, inactivated COVID-19 vaccination could increase the risk of Omicron infection (OR = 1.942, 95% CI: 1.093–3.448), and the receipt of at least one dose of non-inactivated COVID-19 vaccines was a protective factor against infection (OR = 0.428, 95% CI: 0.226–0.812). By contrast, no relations were observed in COVID-19 vaccination with the symptoms of infection and duration of symptoms (*p* > 0.05). Conclusions: This cross-sectional study concluded that inactivated COVID-19 vaccination might increase the risk of Omicron infection, which should be a concern during COVID-19 vaccination and the treatment of variant infections in the future, and the receipt of at least one dose of non-inactivated COVID-19 vaccine was a protective factor against infection.

## 1. Introduction

The COVID-19 global pandemic has been declared to be over [1], while the pathogen, severe acute respiratory syndrome coronavirus 2 (SARS-CoV-2), continues to evolve, and outbreaks of COVID-19 continue to pose a threat.

Various vaccines have been developed to fight against this pandemic, including inactivated vaccines (e.g., BBIBP-CorV, WIBP-CorV and CoronaVac), adenovirus vector vaccines (CanSino Ad5-nCoV) and recombinant subunit vaccines (ZF2001). Clinical trials have shown that these vaccines had good protective effects during the early epidemic [2,3,4,5]. Nevertheless, the significant mutation in the spike protein of variants induces increased infectivity and immune evasion [6]. With the decay of neutralizing antibodies over time and the emergence of new variants, vaccines have offered only limited protection for symptomatic disease caused by the Omicron variant [7].

Furthermore, sub-neutralizing or non-neutralizing antibodies targeting SARS-CoV-2 emerge in vivo, and previous studies showed that these antibodies tend to increase the intensity of infection [8]. A study in the Netherlands indicated that the risk of becoming infected with the Omicron variant was higher among those who had received a vaccine or been previously infected with other strains of SARS-CoV-2, compared with those who did not have immunity to SARS-CoV-2 [9]. Moreover, several domestic and international in vitro cytologic studies have found that the convalescent serums from infected individuals promote virus entry into cells [10,11,12]. Thus, these studies all suggested that pre-existing poorly neutralizing antibodies in the body could enhance the intensity of SARS-CoV-2 infection. At present, most people have been infected with different variants of SARS-CoV-2 and vaccinated with COVID-19 vaccines. With the emergence of variants, whether the pre-existing antibodies promote infection is an important issue and should be clarified and considered during vaccination and treatment. Therefore, the need to continue to explore the efficacy of COVID-19 vaccines is imperative.

The nationwide COVID-19 vaccination program in China was launched in December 2020 [13], and the majority of the population had completed their COVID-19 vaccinations by December 2022 when a huge wave of infections caused by the Omicron variant rapidly swept across China in almost a month due to modifications in its preventive and control measures in response to the COVID-19 epidemic [14,15]. Therefore, we conducted an online survey via Sojump from 1 January to 15 January 2023 to explore the association between vaccination and infection and evaluate the protective efficacy of COVID-19 vaccination during this pandemic.

## 2. Materials and Methods

### 2.1. Study Design and Population

This cross-sectional study was carried out by the convenience sampling method. We used a self-designed anonymous questionnaire to conduct an online survey from 1 January to 15 January 2023 when the Omicron outbreak was just coming to an end in China, which could provide relatively accurate information. Participants were recruited on sojump (https://www.wjx.cn/ (accessed from 1 January to 15 January 2023), the most commonly used online survey tool in China). We shared links or QR codes with participants via WeChat 8.0.32 (the largest social platform in China) after publishing the questionnaire online. All participants were requested to respond to the inquiries in a voluntary and anonymous manner, having read the informed consent on the front page of the survey. The sample size was estimated based on the formula adapted for the cross-sectional study, *n* = Z_α_^2^ × proportion (1 − proportion)/precision^2^. The threshold of α and precision were set at 0.05, and Z_α_ was 1.96. Based on our preliminary survey, the proportion (prevalence of infection with Omicron) was 82.3%. Therefore, the minimum sample size was 224. During the survey period, a total of 2008 questionnaires were collected. The participants in this study were enlisted without any financial compensation to ensure the validity of the findings.

### 2.2. Questionnaires

The questionnaire comprised three parts. The first part collected demographic information, including age, gender, occupation, history of underlying disease and history of COVID-19 vaccination. The second part investigated vaccination status against SARS-CoV-2 and infection status, such as the types and status of COVID-19 vaccines (the questionnaire can be linked directly to ‘Yueshengshi’, the official vaccination information record platform), infected frequency and the prevalence of infection with Omicron. We determined infection outcomes based on positive nucleic acid or antigen tests, or the onset of symptoms related to COVID-19 and ascertained whether the infecting strain was the Omicron variant according to monitoring results from the Chinese Center for Disease Control and Prevention [14]. In the third part, we surveyed the symptoms of infected patients and the duration of symptoms. We devised up to two questions on symptom severity and the duration of symptoms to assess the 12 symptoms in infected patients, which included fever, hypodynamia, muscular soreness, cough and expectoration, nasal congestion and runny nose, pharyngalgia, dental ulcer, dyspnea, nausea and vomiting, diarrhea, encephalalgia and loss of taste or smell. We assigned no clinical symptoms a score of 0 and increased this by one score for each higher level. Finally, the total score for 12 symptoms was calculated, in which the highest score was 36 points, and the lowest score was 0. Symptom severity was classified into two levels, mild and severe, based on the median score of 12 points. We defined a score greater than or equal to 12 points as severe and a score of less than 12 points as mild.

The validation of this questionnaire was mainly examined and approved by professors and relevant professionals before being distributed. Reliability analysis showed that, in the questionnaire, the Cronbach’s alpha of symptom severity was 0.745, which was greater than 0.7, indicating that the questionnaire had good reliability. Additionally, informed consent was obtained from each participant.

### 2.3. Statistical Analysis

All data were analyzed by SPSS Statistics 25.0 software (SPSS Inc., Chicago, IL, USA). Categorical variables were compared by chi-squared test or Fisher’s exact test. Excluding one invalid questionnaire with lack of age information, a total of 2007 questionnaires were ultimately included in the descriptive statistical analysis. Excluding suspected cases (only relevant symptoms) and samples vaccinated after the infection, as well as those with lack of vaccination information, 1434 samples were included in the binary logistic regression analysis of COVID-19 vaccination and infection outcomes. Next, we excluded 576 samples with incomplete vaccination date (missing months or days) and 858 samples were further included in the binary logistic regression analysis of COVID-19 vaccination and infection outcomes. In addition, we excluded the uninfected population and samples with lack of duration of symptom information in 1434 participants, and 986 samples included in the binary logistic regression analysis of COVID-19 vaccination and symptom severity, and ordinal logistic regression analysis of COVID-19 vaccination and the duration of symptoms (Figure 1). Factors with *p* < 0.1 in the univariate logistic regression analysis were included in the final multiple logistic regression analysis, and a significant difference was set at *p* < 0.05. Odds ratios (ORs) and 95% confidence intervals (CIs) were used to estimate their associations.

## 3. Results

### 3.1. Descriptive Statistics for Infection with SARS-CoV-2

During the survey period, a total of 2008 questionnaires were collected. After screening, 2007 valid questionnaires were finally analyzed in this study. Among them, the prevalence of infection with Omicron was 82.9%, which indicated that 1663 had already contracted SARS-CoV-2, with 344 (17.1%) not infected yet. The age of participants ranged from 4 to 85, with a median age of 37 years old and the largest proportion between 18 and 39 (53.6%). Furthermore, respondents had a wide range of occupations, among which students (23.7%), hospital and CDC staffs (18.4%), company employees (14.1%) and staff of government and the public sector (12.2%) accounted for a large proportion. Meanwhile, 85.5% of the participants had no underlying disease previously, and 96.3% of them were vaccinated against SARS-CoV-2, with only 3.7% unvaccinated. Finally, the results showed statistically significant differences in age, occupation and history of COVID-19 vaccination with participants’ infection outcomes (Table 1).

The results of the epidemic curve showed that cases started to appear on 1 December 2022, and subsequently gradually increased, reaching a peak on 20 December 2022 and presenting a downward trend from then on (Figure 2). Based on the number of residents living with each participant, including the infected cases, the average secondary attack rate (SAR) was calculated to be 91.7%. Of the 1663 infected patients, only 175 (10.5%) chose to seek medical care. Regarding the first symptoms and the symptoms during the whole course of infection in patients, the distributions were slightly different, among which fever was the most common, accounting for 20.8% and 15.0%, respectively (Figure 3).

### 3.2. Binary Regression Analysis of COVID-19 Vaccination and Infection Outcomes

To further investigate the relationship between COVID-19 vaccination and infection outcomes, we excluded suspected cases (only those with relevant symptoms but without nucleic acid or antigen testing), and samples who were vaccinated after the infection, as well as those with lack of vaccination information. Finally, a total of 1434 samples were included in the analysis. Out of the 1434 participants, 1139 (79.4%) were infected, whereas 295 (20.6%) remained uninfected. Among them, 592 (41.3%) were male, and 842 (58.7%) were female. The age of participants ranged from 4 to 83, with a median age of 37. COVID-19 vaccination was classified into three types: unvaccinated, vaccinated only with inactivated vaccines and vaccinated with at least one dose of non-inactivated vaccines (adenovirus vector vaccine, recombinant protein vaccines, or mRNA vaccines).

We firstly conducted univariable logistic regression to evaluate the association among vaccination status, sociodemographic characteristics and infection outcome of SARS-CoV-2. The results of logistic regression analysis showed that type of vaccination (*p* < 0.001) and occupation (*p* = 0.004) were associated with the infection outcome of SARS-CoV-2 (Table 2). These factors were further subjected to multiple logistic regression. After adjustment, compared with unvaccinated people, participants who were vaccinated only with inactivated vaccines were more likely to be infected (OR = 1.942, 95% CI: 1.093–3.448), whereas those who had received at least one dose of non-inactivated vaccine were at less risk from Omicron infection (OR = 0.428, 95% CI: 0.226–0.812). The adjusted model passed the Hosmer and Lemeshow test (χ^2^ = 1.162, df = 6, *p* = 0.979), indicating that this model was a good fit. The results of the collinearity test showed that tolerance > 0.5 and VIF < 3, which suggested no collinearity among independent variables (Figure 4).

The time interval between the last vaccination and infection could be a potential factor affecting the infection rate of SARS-CoV-2. To further assess the impact of time interval on regression results, we excluded 576 samples with incomplete vaccination date (missing months or days) and 858 samples were further included in the binary logistic regression analysis of COVID-19 vaccination and infection outcomes. The results did not change substantially compared to the previous regression analysis (Table 3). Compared with unvaccinated people, participants who were vaccinated only with in-activated vaccines were more likely to be infected (OR = 1.889, 95% CI: 1.069–3.338), whereas those who had received at least one dose of non-inactivated vaccine were at less risk from Omicron infection (OR = 0.468, 95% CI: 0.244–0.895).


### 3.3. Regression Analysis of COVID-19 Vaccination and Symptom Severity or Duration of Symptoms

To further explore the association between severity of symptoms or duration of symptoms with COVID-19 vaccination, we excluded the uninfected population and samples with a lack of duration of symptom information. We finally included 986 population samples for regression analysis. Of the 986 participants, 419 (42.50%) were male, and 567 (57.5%) were female. The ages of participants, which spanned from 4 to 83, were 37 on a median.

We first conducted univariable logistic regression to evaluate the association between vaccination status, sociodemographic characteristics and symptom severity. The results of logistic regression analysis showed that five factors, namely, type of vaccination (*p* = 0.044), vaccination status (*p* = 0.037), gender (*p* < 0.001), age (*p* < 0.001) and occupation (*p* = 0.002), were associated with symptom severity (Table 4). These factors were included in the multiple logistic regression. After adjustment, the results showed no statistically significant differences in COVID-19 vaccination with symptom severity (Figure 5). The adjusted model passed the Hosmer and Lemeshow test (χ^2^ = 13.669, df = 8, *p* > 0.05), indicating that this model was a good fit. The result of the collinearity test showed that the tolerance > 0.5 and VIF < 3, which suggested no collinearity among independent variables.

We continued to explore the relationship between the duration of symptoms and COVID-19 vaccination via univariable logistic regression. The results of the regression analyses showed no statistically significant differences in type of vaccination and vaccination status with the duration of symptoms (Table 5).

In summary, we found no association between COVID-19 vaccination and the symptom severity or duration of symptoms.

## 4. Discussion

In this cross-sectional study, we found that the receipt of at least one dose of a non-inactivated COVID-19 vaccine was a protective factor against infection. Non-inactivated COVID-19 vaccines could offer improved protective efficacies compared with inactivated vaccines, and this could be related to the time of vaccination. As early as 12 October 2020, appointments for COVID-19 vaccination began in Beijing and Wuhan [16]. Inactivated vaccines were the earliest vaccines provided for large-scale vaccination in China. The immunity provided by vaccines waned over time. Therefore, protection effectiveness was not as strong as that of other types of vaccines [17]. In addition, T-cell immunity is also an essential factor in the effectiveness of vaccines. T cells have a significant role in the enduring immunity provided by immune memory [18]. Compared with other types of vaccines, the inactivated vaccine had a considerably reduced induction of spike-specific IFN-γ secreting T-cell [19], which plays a crucial part in the fight against SARS-CoV-2 infection [20,21].

We also found that inactivated COVID-19 vaccination was a risk factor for infection, which was consistent with the previous research findings. A study in the Netherlands [9] suggested that the protective effectiveness of different variant-originated COVID-19 vaccines is inadequate, and receiving the vaccine may increase the chance of being infected with Omicron BA.1. Similarly, our previous research also found that vaccination actually increased the risk of becoming infected by Omicron, compared to those who had not been vaccinated, during the Omicron outbreak in Guangzhou in May 2022 [22]. Furthermore, a number of in vitro cytological experiments have found that convalescent serums from infected individuals promote new variants’ entry into cells [10,11,12]. This could be attributed to the fact that inactivated vaccines were developed based on the original strain of SARS-CoV-2. The antibodies produced in the body waned over time, which resulted in poorly neutralizing antibodies for new variants like Omicron. Therefore, they could not effectively neutralize the virus and instead enhanced the intensity of infection, which was consistent with previous research observations [8]. On the other hand, it is possible that unvaccinated individuals may have implemented strict protection measures during the pandemic, potentially leading to the adverse association between vaccination and infection.

Additionally, we found that, the older the age, the milder the symptoms after infection. This may be due to the decline in physiological functions with aging, which might result in a “delayed response” to the invasion of SARS-CoV-2. After being infected with the virus, there is usually no high fever, runny nose, or other typical prominent symptom [23]. Therefore, it is necessary to carefully handle the atypical clinical symptoms of elderly COVID-19 patients. On the other hand, we found that, compared with males, females would experience more severe symptoms after infection. However, current research has found that adult females could produce more interferon-α (IFNα) from plasmacytoid dendritic cells compared to adult males [24], which provided females with a certain level of protection. This was not consistent with our research findings, which may be due to the fact that this study was a cross-sectional study, with a higher number of female participants than male participants, resulting in an imbalance in baseline characteristics and potential bias.

This study has strengths and limitations. Firstly, given the vaccination strategy for the COVID-19 vaccine and strict preventive and control measures in China, the time of vaccination in most of the population occurred before SARS-CoV-2 infection, benefiting the inference of cause and effect. Secondly, as a result of changes in China’s prevention and control policies, healthcare facilities will no longer proactively provide nucleic acid or antigen testing for suspected infections, and there is no way to accurately identify infected and non-infected individuals in China. However, the study included population samples that could effectively identify infected and non-infected individuals, avoiding bias and allowing an accurate exploration of the association between COVID-19 vaccination and SARS-CoV-2 infection. Finally, the majority of the questionnaire sample was from Guangdong, which limited extrapolation from the results. However, it can still provide a preliminary basis for the study of the association between vaccination and infection.

## 5. Conclusions

This study concluded that inactivated COVID-19 vaccination might increase the risk of SARS-CoV-2 Omicron variant infection, which should be taken into consideration during COVID-19 vaccination and when treating future variants of the infection. The receipt of at least one dose of non-inactivated COVID-19 vaccines was a protective factor against infection. In future, COVID-19 vaccination should be optimized to constantly adjust and explore the optimal vaccination regimens.

## Figures and Tables

**Figure 1 vaccines-12-00567-f001:**
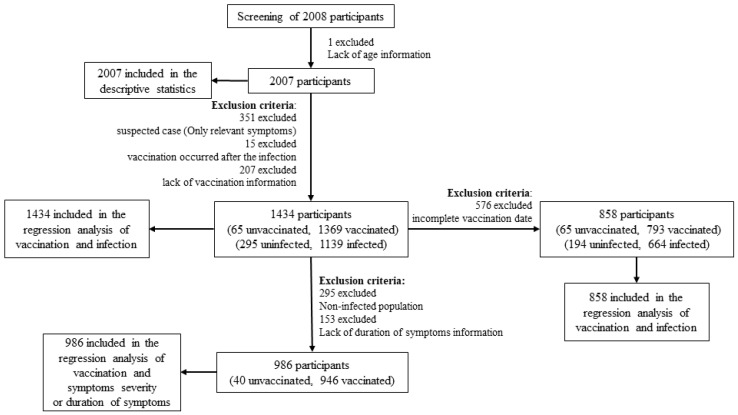
Flow chart of statistical analysis.

**Figure 2 vaccines-12-00567-f002:**
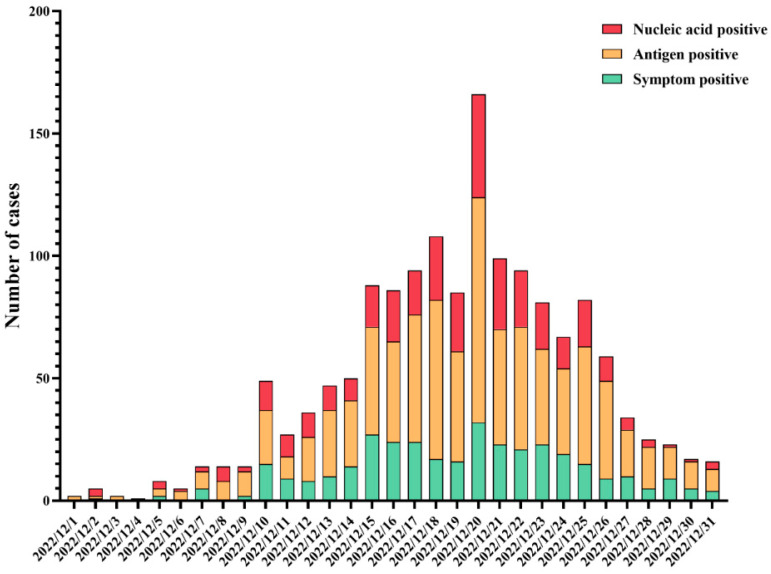
Epidemic curve of SRAS-CoV-2 infections in December 2022.

**Figure 3 vaccines-12-00567-f003:**
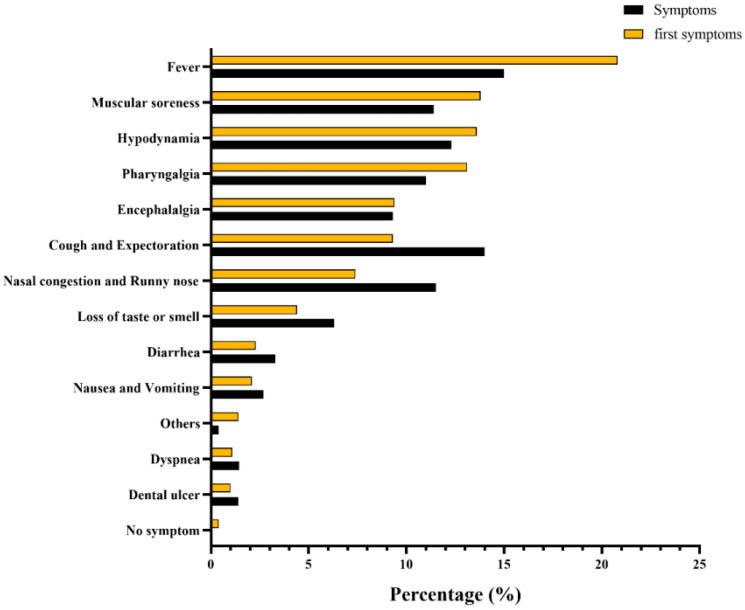
Frequency distribution of first symptoms and symptoms of SARS-CoV-2 infection.

**Figure 4 vaccines-12-00567-f004:**
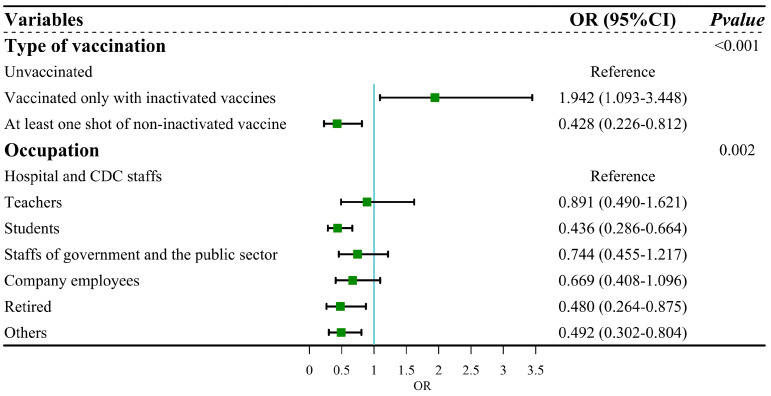
Forest plot: multiple regression for infection outcome of SARS-CoV-2.

**Figure 5 vaccines-12-00567-f005:**
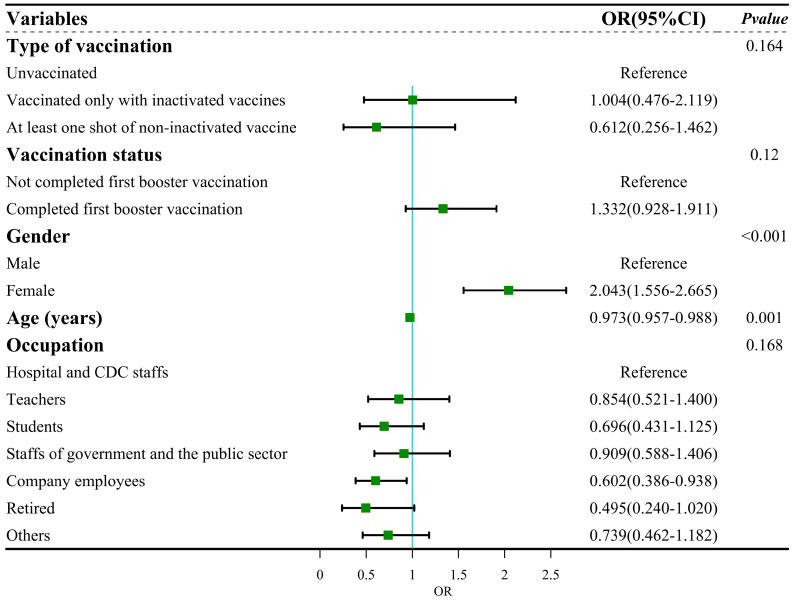
Forest plot: multiple regression for symptom severity.

**Table 1 vaccines-12-00567-t001:** Distribution of the demographic characteristics of the participants (n = 2007).

Variables	Total Participants	Infection Outcome		*p* Value (χ^2^ Test of Fisher’s Exact Test)	OR (95% CI) (Univariable Regression Analysis)	*p* Value (Univariable Regression Analysis)
	(*n*/%) (*n* = 2007)	Infected (*n*/%) (*n* = 1663)	Uninfected (*n*/%) (*n* = 344)			
**Gender**				0.732		0.732
Male	841 (41.9)	694 (41.7)	147 (42.7)		0.96 (0.759–1.214)	
Female	1166 (58.1)	969 (58.3)	197 (57.3)		Reference	
**Age (years)**				0.001		0.002
Less than 18	23 (1.1)	16 (1.0)	7 (2.0)		1.000 (0.370–2.699)	
18–39	1076 (53.6)	896 (53.9)	180 (52.3)		2.178 (1.358–3.492)	
40–59	816 (40.7)	687 (41.3)	129 (37.5)		2.330 (1.438–3.774)	
60 and older	92 (4.6)	64 (3.8)	28 (8.1)		Reference	
**Occupation**				0.001		0.001
Hospital and CDC staffs	369 (18.4)	318 (19.1)	51 (14.8)		Reference	
Teachers	205 (10.2)	184 (11.1)	21 (6.1)		1.405 (0.819–2.410)	
Students	475 (23.7)	376 (22.6)	99 (28.8)		0.609 (0.421–0.881)	
Staffs of government and the public sector	245 (12.2)	204 (12.3)	41 (11.9)		0.798 (0.510–1.248)	
Company employees	282 (14.1)	242 (14.6)	40 (11.6)		0.970 (0.621–1.516)	
Retired	131 (6.5)	99 (6.0)	32 (9.3)		0.496 (0.302–0.815)	
Others	300 (14.9)	240 (14.4)	60 (17.4)		0.642 (0.426–0.966)	
**History of underlying disease**				0.786		0.787
No prior medical history	1716 (85.5)	1421 (85.4)	295 (85.8)		Reference	
Cerebral-cardiovascular diseases	146 (7.3)	124 (7.5)	22 (6.4)		1.170 (0.731–1.873)	
Pulmonary disease	39 (1.9)	33 (2.0)	6 (1.7)		1.142 (0.474–2.750)	
Others	106 (5.3)	85 (5.1)	21 (6.1)		0.840 (0.513–1.377)	
**History of vaccination**				0.047		0.050
Unvaccinated	74 (3.7)	55 (3.3)	19 (5.5)		Reference	
Vaccinated	1933 (96.3)	1608 (96.7)	325 (94.5)		1.709 (1.001–2.918)	

**Table 2 vaccines-12-00567-t002:** Univariable logistic regression analysis of influencing factors for infection outcome of SARS-CoV-2 (n = 1434).

Variables	Dependent Variable		β	Odds Ratio	95% CI	*p* Value
	Infected (*n*/%)	Uninfected (*n*/%)				
**Type of vaccination**						<0.001
Unvaccinated	46 (4.0)	19 (6.4)	—	Reference		
Vaccinated only with inactivated vaccines	1007 (88.4)	203 (68.8)	0.717	2.049	1.176–3.570	
At least one shot of non-inactivated vaccine	86 (7.6)	73 (24.7)	−0.720	0.487	0.262–0.904	
**Vaccination status**						0.553
Not completed first booster vaccination	222 (19.5)	53 (18.0)	—	Reference		
Completed first booster vaccination	917 (80.5)	242 (82.0)	−0.100	0.905	0.649–1.260	
**Gender**						0.997
Male	470 (41.3)	122 (41.4)	—	Reference		
Female	669 (58.7)	173 (58.6)	0.004	1.004	0.774–1.302	
**Age (years)**	—	—	−0.001	0.999	0.989–1.009	0.871
**Occupation**						0.004
Hospital and CDC staffs	241 (21.2)	45 (15.3)	—	Reference		
Teachers	116 (10.2)	19 (6.4)	0.131	1.140	0.638–2.036	
Students	251 (22.0)	87 (29.5)	−0.619	0.539	0.361–0.804	
Staffs of government and the public sector	170 (14.9)	38 (12.9)	−0.180	0.835	0.520–1.342	
Company employees	168 (14.7)	38 (12.9)	−0.192	0.826	0.514–1.327	
Retired	61 (5.4)	24 (8.1)	−0.745	0.475	0.269–0.839	
Others	132 (11.6)	44 (3.1)	−0.580	0.560	0.351–0.893	
**History of underlying disease**						0.591
No prior medical history	966 (84.8)	256 (86.8)	—	Reference		
Cerebral-cardiovascular diseases	86 (7.6)	16 (5.4)	0.354	1.424	0.821–2.472	
Pulmonary disease	24 (2.1)	5 (1.7)	0.241	1.272	0.481–3.367	
Others	63 (5.5)	18 (6.1)	−0.75	0.928	0.540–1.594	

**Table 3 vaccines-12-00567-t003:** Univariable logistic regression analysis of influencing factors for infection outcome of SARS-CoV-2 (n = 858).

Variables	Dependent Variable		β	Odds Ratio	95% CI	*p* Value
	Infected (*n*/%)	Uninfected (*n*/%)				
**Type of vaccination**						<0.001
Unvaccinated	46 (6.9)	19 (9.8)	—	Reference		
Vaccinated only with inactivated vaccines	558 (84.0)	122 (62.9)	0.639	1.889	1.069–3.338	
At least one shot of non-inactivated vaccine	60 (9.0)	53 (27.3)	−0.760	0.468	0.244–0.895	
**Vaccination status**						0.619
Not completed first booster vaccination	141 (21.2)	38 (19.6)	—	Reference		
Completed first booster vaccination	523 (78.8)	156 (80.4)	−0.101	0.904	0.605–1.348	
**Time interval**						0.352
Unvaccinated	46 (6.9)	19 (9.8)		Reference		
<1 years	344 (51.8)	93 (47.9)	0.424	1.528	0.854–2.733	
>1 years	274 (41.3)	82 (23.0)	0.322	1.380	0.766–2.487	
**Gender**						0.634
Male	251 (37.8)	77 (39.7)	—	Reference		
Female	413 (77.9)	117 (60.3)	0.080	1.083	0.780–1.503	
**Age (years)**	—	—	−0.005	0.995	0.983–1.007	0.422
**Occupation**						0.339
Hospital and CDC staffs	132 (19.9)	35 (18.0)	—	Reference		
Teachers	67 (10.1)	12 (6.2)	0.392	1.480	0.722–3.037	
Students	142 (21.4)	48 (24.7)	−0.243	0.784	0.478–1.288	
Staffs of government and the public sector	104 (15.1)	27 (13.9)	0.021	1.021	0.581–1.795	
Company employees	100 (15.1)	26 (13.4)	0.020	1.020	0.577–1.803	
Retired	50 (7.5)	21 (10.8)	−0.460	0.631	0.366–1.187	
Others	69 (10.4)	25 (12.9)	−0.312	0.732	0.406–1.320	
**History of underlying disease**						0.366
No prior medical history	545 (76.4)	168 (86.6)	—	Reference		
Cerebral-cardiovascular diseases	59 (8.9)	12 (6.2)	0.416	1.516	0.796–2.887	
Pulmonary disease	17 (2.6)	2 (1.0)	0.963	2.620	0.599–11.457	
Others	43 (6.5)	12 (6.2)	0.099	1.105	0.569–2.143	

**Note:** The time interval refers to the period from the last vaccination to the infection; If no infection has occurred, the time interval is calculated based on the submission time of the questionnaire. Since only the type of vaccination had statistical significance, the results of multiple logistic regression were consistent with univariable logistic regression.

**Table 4 vaccines-12-00567-t004:** Univariable logistic regression analysis of influencing factors for symptom severity (n = 986).

Variables	Dependent Variable		β	Odds Ratio	95% CI	*p* Value
	Mild (*n*/%)	Severe (*n*/%)				
**Type of vaccination**						0.044
Unvaccinated	25 (5.2)	15 (3.0)	—	Reference		
Vaccinated only with inactivated vaccines	418 (86.2)	457 (91.2)	0.600	1.822	0.948–3.503	
At least one shot of non-inactivated vaccine	42 (8.7)	29 (5.8)	0.140	1.151	0.519–2.551	
**Vaccination status**						0.037
Not completed first booster vaccination	110 (22.7)	87 (17.4)	—	Reference		
Completed first booster vaccination	375 (77.3)	414 (82.6)	0.334	1.396	1.020–1.911	
**Gender**						<0.001
Male	246 (50.7)	173 (34.5)	—	Reference		
Female	239 (49.3)	328 (65.5)	0.669	1.951	1.511–2.521	
**Age (years)**	**—**	**—**	−0.027	0.973	0.963–0.984	<0.001
**Occupation**						0.002
Hospital and CDC staffs	91 (18.8)	115 (23.0)	—	Reference	—	
Teachers	51 (10.5)	51 (10.2)	−0.234	0.791	0.492–1.273	
Students	85 (17.5)	122 (24.4)	0.127	1.136	0.769–1.678	
Staffs of government and the public sector	73 (15.1)	77 (15.4)	−0.181	0.835	0.547–1.273	
Company employees	83 (17.1)	63 (12.6)	−0.510	0.601	0.392–0.921	
Retired	39 (8.0)	17 (3.4)	−1.064	0.345	0.183–0.649	
Others	63 (13.0)	56 (11.2)	−0.352	0.703	0.447–1.106	
**History of underlying disease**						0.123
No prior medical history	404 (83.3)	433 (86.4)	—	Reference	—	
Cerebral-cardiovascular diseases	48 (9.9)	31 (6.2)	−0.507	0.603	0.376–0.966	
Pulmonary disease	6 (1.2)	11 (2.2)	0.537	1.711	0.627–4.668	
Others	27 (5.6)	26 (5.2)	−0.107	0.898	0.516–1.566	

**Table 5 vaccines-12-00567-t005:** Univariable logistic regression analysis of influencing factors for the duration of disease (n = 986).

Variables	Dependent Variable (Days)			β	Odds Ratio	95% CI	*p* Value
	Less than 3 (*n*/%)	4–6 (*n*/%)	7 and More (*n*/%)				
**Type of vaccination**							
Unvaccinated	11 (6.9)	11 (2.7)	18 (4.2)	—	Reference	—	—
Vaccinated only with inactivated vaccines	135 (84.9)	358 (89.3)	382 (89.7)	0.222	1.249	0.691–2.259	0.462
At least one shot of non-inactivated vaccine	13 (8.2)	32 (8.0)	26 (6.1)	−0.039	0.962	0.466–1.984	0.915
**Vaccination status**							
Not completed first booster vaccination	30 (18.9)	80 (20.0)	87 (20.4)	—	Reference	—	—
Completed first booster vaccination	129 (81.1)	321 (80.0)	339 (79.6)	−0.058	0.944	0.703–1.266	0.699
**Age (years)**	—	—	—	0.018	1.018	1.008–1.028	<0.001
**Occupation**							
Hospital and CDC staffs	17 (10.7)	92 (22.9)	97 (22.8)	—	Reference	—	—
Teachers	14 (8.8)	39 (9.7)	49 (11.5)	−0.070	0.932	0.593–1.465	0.761
Students	39 (24.5)	96 (23.9)	72 (16.9)	−0.564	0.569	0.395–0.820	0.002
Staffs of government and the public sector	19 (11.9)	64 (16.0)	67 (15.7)	−0.154	0.857	0.575–1.279	0.451
Company employees	31 (19.5)	63 (15.7)	52 (12.2)	−0.594	0.552	0.370–0.824	0.004
Retired	10 (6.3)	11 (2.7)	35 (8.2)	0.360	1.433	0.803–2.563	0.224
Others	29 (18.2)	36 (9.0)	54 (12.7)	−0.376	0.687	0.448–1.051	0.084
**History of underlying disease**							
No prior medical history	135 (84.9)	355 (88.5)	347 (81.5)	—	Reference	—	—
Cerebral-cardiovascular diseases	13 (8.2)	26 (6.5)	40 (9.4)	0.276	1.318	0.850–2.044	0.217
Pulmonary disease	3 (1.9)	4 (1.0)	10 (2.3)	0.538	1.713	0.672–4.362	0.260
Others	8 (5.0)	16 (4.0)	29 (6.8)	0.439	1.551	0.908–2.649	0.108

## Data Availability

The datasets used and/or analyzed during the current study are available from the corresponding author on reasonable request.

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
