# Peer review of "Have Previous COVID-19 Vaccinations Shaped the Potential Enhancing Infection of Variant Strains?"

_vaccines, 2024, doi:10.3390/vaccines12060567_

Round 1
Reviewer 1 Report
Comments and Suggestions for Authors
The authors have submitted the manuscript entitled “Have previous COVID-19 vaccinations shaped the potential enhancing infection of variant strains?” In this, the authors evaluate the relationship between the Omicron infection and the receipt of COVID-19 vaccination.
Overall, it is a well-designed study. However, I have some concerns that need to be addressed:
1. How do the authors classify the SARS-COV-2 infection as Omicron vs. non-Omicron infection? Is it PCR-confirmed or purely interpreted based on the time of infection and the main circulating variant at the time?
2. In Figure 1, the authors write that 1 participant was excluded based on the lack of age information. However, in line 88 of the manuscript, they mention that the participant had erroneous age information.
3. How do the authors classify previously infected vs. non-infected individuals? In the case of self-reporting, have they into consideration the asymptomatic infections (especially in the case of health care workers)?
4. In lines 177-178 of the manuscript, the authors should mention that they have included the uninfected population in this analysis as well. The authors write that “we included only confirmed cases of infection (nucleic acid-positive and antigen-positive) from the infected population and excluded samples who were vaccinated after infection”. This makes the reader mistakenly believe that the uninfected population was not included in the analysis. This should be clarified.
5. The time interval between the last dose of the vaccine (dose 2 of the primary series or the booster vaccination) and the infection must be taken into consideration. Without this, the conclusion of ADE seems far-fetched. The increased rate of infection could just be attributed to the waning immunity to the vaccines as opposed to the ADE effect. As the authors mentioned in lines 265-266, the inactivated vaccines were the earliest available vaccines in China. Thus, the risk of Omicron infection might as well arise from the waning of the vaccine-induced immunity instead of the ADE effect. This must be accounted for before concluding the ADE effect.
Author Response
The authors have submitted the manuscript entitled “Have previous COVID-19 vaccinations shaped the potential enhancing infection of variant strains?” In this, the authors evaluate the relationship between the Omicron infection and the receipt of COVID-19 vaccination.
Overall, it is a well-designed study. However, I have some concerns that need to be addressed:
- How do the authors classify the SARS-COV-2 infection as Omicron vs. non-Omicron infection? Is it PCR-confirmed or purely interpreted based on the time of infection and the main circulating variant at the time?
Response: Thank you for your further comments. The inclusion of our research subjects is based on whether they are infected with the SARS-CoV-2 through nucleic acid testing or antigen testing. As to whether it is the Omicron infection, according to the monitoring report of the Chinese Center for Disease Control and Prevention. The report showed that from September 26, 2022, to April 6, 2023, a total of 38,655 cases of local COVID-19 infections have been reported nationwide with valid genome sequences, all of which were Omicron variant strains.
(https://www.chinacdc.cn/jkzt/crb/zl/szkb_11803/jszl_13141/202304/t20230408_264979.html)
- In Figure 1, the authors write that 1 participant was excluded based on the lack of age information. However, in line 88 of the manuscript, they mention that the participant had erroneous age information.
Response: Thanks for your comments. The wording has been adjusted in the text. However, for consistency with the context, the description of sample selection has been moved to the statistical analysis section (Line 117-119).
- How do the authors classify previously infected vs. non-infected individuals? In the case of self-reporting, have they into consideration the asymptomatic infections (especially in the case of health care workers)?
Response: Thanks for your comments. This is a very constructive suggestion. Prior to this, China implemented strict epidemic prevention and control policies, employing large-scale nucleic acid testing to ensure that all individuals underwent testing every 24 hours, which was highly reliable for determining infection outcomes, including previously infected and non-infected individuals as you mentioned. Thought some people without any symptom may choose not to undergo relevant testing since large-scale nucleic acid testing has not been conducted, during the peak of the pandemic, driven by the current epidemiological situation and the infection status of close contacts, the majority of individuals spontaneously undergo nucleic acid or antigen testing, maximizing the identification of asymptomatic infections. What’s more, for certain high-risk occupations, especially health care workers, daily nucleic acid testing was required.
- In lines 177-178 of the manuscript, the authors should mention that they have included the uninfected population in this analysis as well. The authors write that “we included only confirmed cases of infection (nucleic acid-positive and antigen-positive) from the infected population and excluded samples who were vaccinated after infection”. This makes the reader mistakenly believe that the uninfected population was not included in the analysis. This should be clarified.
Response: Thank you for your further comments. We have revised the relevant statement in the article (Line 180-183).
- The time interval between the last dose of the vaccine (dose 2 of the primary series or the booster vaccination) and the infection must be taken into consideration. Without this, the conclusion of ADE seems far-fetched. The increased rate of infection could just be attributed to the waning immunity to the vaccines as opposed to the ADE effect. As the authors mentioned in lines 265-266, the inactivated vaccines were the earliest available vaccines in China. Thus, the risk of Omicron infection might as well arise from the waning of the vaccine-induced immunity instead of the ADE effect. This must be accounted for before concluding the ADE effect.
Response: Thank you for this valuable comments. In fact, even as the antibody levels in the bodies of vaccinated individuals decrease over time, their infection probability should not be higher than that of unvaccinated individuals. However, current results indicate that the risk of infection is higher in the population vaccinated with all inactivated vaccines compared to the unvaccinated. In this situation, we cannot rule out the possibility of potential ADE effects. Most importantly, we strongly hold that the time interval between the last dose of the vaccine and the infection is a very important factor that needs to be considered. We have added relevant analyses and statements in the article (Line122-124, Line 202-211, Table3). The results indicated that there was no statistically significant association between the time interval and infection. On the one hand, due to the policy of vaccination, people's vaccination schedules tend to be similar in China. On the other hand, due to the changes in China's prevention and control policies, people were mostly infected with the virus in December 2022. Therefore, these led to small differences in the time intervals among the population.

Reviewer 2 Report
Comments and Suggestions for Authors
Estimated Authors,
I've read with great interest the present cross-sectional study based on an internet-delivered questionnaire regarding vaccine acceptance, infection by SARS-CoV-2 and symptoms complained by a sample of around 2000 people from Mainland China.
According to the present study, infection was associated with being vaccinated with inactivated vaccines (OR 2.049; 95%CI 1.176 to 3.570), while a mild disease severity compared to severe disorder was associated with a completed first booster vaccination (OR 1.396; 95%CI 1.020 to 1.911), male gender (OR 1.951; 95%CI 1.511 to 2.521) with a protective role of age (OR 0.973, 95%CI 0.963 to 0.984), having a history of cardiovascular disease (OR 0.603, 95%CI 0.376 to 0.966 - a possible role of having a more rigourous seclusion? please discuss). Interestingly, students exhibited some specificities, possibly associated with low age at the time of the pandemic.
The paper is not particularly innovative, as otherwise acknowledged by study Authors, but also interesting for being based on mainland China, where the zero-covid policy has been implemented longer than otherwise in highly developed countries, and where vaccination campaigns have been initially performed by means of an inactivated formulate that has been otherwise not employed in most of highly developed countries.
However, Authors are welcome to adjust some uncertainties in the reporting of their data.
More precisely:
1) Flow chart is unclear and not properly reporting the flow of cases: at the first step, the graphical option implemented by Authors suggests that 2007 students were removed from the analysis - while all of them clearly progressed in the following step. Similarly, at the second step for the descriptive analysis, while Authors do not report how 2007 entries were reduced to 1434, and so on.
2) ADE is a specifically defined medical condition: Authors have properly inquired the disease severity as reported by participants, but the study is not designed in order to ascertain whether any of participants did experience or not ADE. Therefore, the use of this term may be misleading and confusing, and should be avoided.
3) The size of the sample should be included only in results; the statement from row 87 is therefore confusing and misleading, please remove.
Author Response
Response to Reviewer 2’s Comments
I've read with great interest the present cross-sectional study based on an internet-delivered questionnaire regarding vaccine acceptance, infection by SARS-CoV-2 and symptoms complained by a sample of around 2000 people from Mainland China.
According to the present study, infection was associated with being vaccinated with inactivated vaccines (OR 2.049; 95%CI 1.176 to 3.570), while a mild disease severity compared to severe disorder was associated with a completed first booster vaccination (OR 1.396; 95%CI 1.020 to 1.911), male gender (OR 1.951; 95%CI 1.511 to 2.521) with a protective role of age (OR 0.973, 95%CI 0.963 to 0.984), having a history of cardiovascular disease (OR 0.603, 95%CI 0.376 to 0.966 - a possible role of having a more rigourous seclusion? please discuss). Interestingly, students exhibited some specificities, possibly associated with low age at the time of the pandemic.
Response: Thank you for your further comments. In univariable logistic regression analysis, the results showed that five factors, namely, type of vaccination (P=0.044), vaccination status (P=0.037), gender (P<0.001), age (P<0.001) and occupation (P=0.002), were associated with symptom severity. However, only age and gender were statistically significant in relation to symptom severity in multiple regression analysis. We hold that the results of multiple regression analysis are more reliable compared to univariable logistic regression analysis. What’s more, relevant analysis and discussion have been added to the discussion section of this paper (Line 308-320).
The paper is not particularly innovative, as otherwise acknowledged by study Authors, but also interesting for being based on mainland China, where the zero-covid policy has been implemented longer than otherwise in highly developed countries, and where vaccination campaigns have been initially performed by means of an inactivated formulate that has been otherwise not employed in most of highly developed countries.
However, Authors are welcome to adjust some uncertainties in the reporting of their data.
Response: Thank you for your comments. I understand your concerns about the accuracy of the results. I need to explain to you the efforts we have made in data processing to ensure the accuracy of the results. First, we determined infected and uninfected based on positive nucleic acid, or positive antigen, or the presence of associated symptoms. However, in order to avoid selection bias, the samples we included in the subsequent analysis excluded suspected cases with only related symptoms. In addition, the data we collected were based on the “Yueshengshi” platform, which is an official platform for recording vaccination information and nucleic acid monitoring results, so as to avoid recall bias in the included population. What’s more, in order to ensure the rigor of causal inference, we carefully compared the time of vaccination with the time of infection when processing the data to ensure that the included samples were vaccinated before the infection. Finally, we also analyzed the effect of the time interval between last vaccination and infection on infection. Relevant analysis has been added to the article (Line122-124, Line 202-211, Table3).
More precisely:
1) Flow chart is unclear and not properly reporting the flow of cases: at the first step, the graphical option implemented by Authors suggests that 2007 students were removed from the analysis - while all of them clearly progressed in the following step. Similarly, at the second step for the descriptive analysis, while Authors do not report how 2007 entries were reduced to 1434, and so on.
Response: Thank you for your comments. We have modified the flow chart in the article (Figure 1).
2) ADE is a specifically defined medical condition: Authors have properly inquired the disease severity as reported by participants, but the study is not designed in order to ascertain whether any of participants did experience or not ADE. Therefore, the use of this term may be misleading and confusing, and should be avoided.
Response: Thank you for your comments. We understand that ADE is a professional term, and its misuse can lead to confusion and misunderstanding. Therefore, we only made relevant statements in the introduction part to highlight the significance of the research, and explained the research results with potential ADE effect in the discussion part, and ADE was not used in the rest part.
3) The size of the sample should be included only in results; the statement from row 87 is therefore confusing and misleading, please remove.
Response: Thank you for your comments. The relevant statement has been removed from the article.

Round 2
Reviewer 2 Report
Comments and Suggestions for Authors
Estimated Authors and Editors,
The paper has been extensively improved according to my previous comments, when possible. With a significant notable exception: the inappropriate use of the term "ADE".
Please understand that, until Authors revise this specific and critical comment I'm unable to change my recommendation and endorse the potential acceptance of this paper - that otherwise I would suggest.
Author Response
Response to Reviewer 2’s Comments
Estimated Authors and Editors,
The paper has been extensively improved according to my previous comments, when possible. With a significant notable exception: the inappropriate use of the term "ADE".
Please understand that, until Authors revise this specific and critical comment I'm unable to change my recommendation and endorse the potential acceptance of this paper - that otherwise I would suggest.
Response: Thank you for your further comments. According to your comments, the mention of ADE has been eliminated from the article, and significant revisions have been applied to the relevant statements (highlighted in the article).

Round 3
Reviewer 2 Report
Comments and Suggestions for Authors
I warmly thank the Authors for their collaborative approach to my requests.
In the end, I'm endorsing the acceptance of the paper.